# Immobilization of Metronidazole on Mesoporous Silica Materials

**DOI:** 10.3390/pharmaceutics14112332

**Published:** 2022-10-29

**Authors:** Klára Szentmihályi, Szilvia Klébert, Zoltán May, Eszter Bódis, Miklós Mohai, László Trif, Tivadar Feczkó, Zoltán Károly

**Affiliations:** Institute of Materials and Environmental Chemistry, Research Centre for Natural Sciences, Magyar Tudósok Körútja 2, H-1117 Budapest, Hungary

**Keywords:** metronidazole, mesoporous silica material, SBA-15, dissolution

## Abstract

Metronidazole (MTZ) is a widely used drug, but due to its many side effects, there is a growing trend today to use a minimum dose while maintaining high efficacy. One way to meet this demand is to reduce the size of the drug particles. A relatively new method of size reduction is attaching the drug molecules to a mesoporous carrier. In this paper, we studied the fixation of MTZ molecules on mesoporous silica carriers. The drug was immobilized on two mesoporous silica materials (Syloid, SBA-15) with the use of a variety of immersion techniques and solvents. The immobilized drug was subjected to physicochemical examinations (e.g., SEM, XPS, XRD, nitrogen uptake, DSC) and dissolution studies. A significantly higher immobilization was attained on SBA-15 than on a Syloid carrier. Among the processing parameters, the type of MTZ solvent had the highest influence on immobilization. Ultrasonic agitation had a lower but still significant impact, while the concentration of MTZ in the solution made no difference. Under optimal conditions, with the application of an ethyl acetate solution, the surface coverage on SBA-15 reached as much as 91%. The immobilized MTZ exhibited a ca. 10% faster dissolution rate as compared to the pure micron-sized drug particles.

## 1. Introduction

Metronidazole (MTZ, 1-[2-hydroxyethyl] -2-methyl-5-nitroimidazole, C_6_H_9_N_3_O_3_, Appendix A) is a widely used systemic and topical drug with bactericidal properties. It has been the most commonly used drug against various protozoa (*Entamoeba histolytica, Giardia lamblia, Trichomonas vaginalis*) as well as Gram-negative (*Bacteroides* and *Fusobacterium* spp.) and Gram-positive anaerobic bacteria (*Peptostreptococcus* and *Clostridia* spp.) since its introduction in 1959 [1,2]. However, its clinical use is constantly growing, and it has now been found to be extremely effective in treating inflammatory bowel diseases such as Crohn’s disease and ulcerative colitis, as well as burns [3,4]. Its in vivo antioxidant and free radical scavenging properties can be attributed to the inactivation of inflammatory mediators released by neutrophils (e.g., reactive oxygen species, IL-8) and the reduction of oxidative stress [5,6]. Although redox homeostasis may be affected by ROS-independent mechanisms based on in vitro studies, its direct antioxidant properties are not supported. Moreover, it acts as a prooxidant in the H_2_O_2_/·OH microperoxidase–luminol system, but is able to inhibit lipid peroxidation [7].

By external application (e.g., in rosacea or in the treatment of acne-prone skin), only its local effect is known, as it is not absorbed through the skin [8,9,10]. However, its applicability and bioavailability are greatly limited by its low water solubility with 10.61 ± 0.51 mg/mL at 25 °C. In high-dose and long-term internal administration, gastrointestinal problems (e.g., gastrointestinal disorders, nausea, abdominal pain and diarrhea) and neurotoxicity may occur [11]. In the case of topical application in cosmetic preparations, e.g., in the treatment of rosacea, even at low concentrations (1% or 0.75%), a number of side effects have been experienced, including dry skin, skin irritation, abnormal redness and inflammation of the skin (erythema), itching, burning sensation, rash, skin pain and hip sensation [12].

For the reduction of these symptoms, it is necessary to reduce the quantity or size of the particles of the active substance, as the size of the drug particles affects a number of properties, including bioavailability, toxicity, shelf life, distribution and efficacy [13,14,15].

There are a number of traditional and new methods for reducing the particle size of poorly water-soluble drugs (micronization, nanonization), such as the melt methods (e.g., melt quenching, hot-melt extrusion, hot-melt granulation, dropping method, direct capsule filling), the solvent methods (e.g., pH-shift method, supercritical fluid method, spray-drying, freeze-drying, spray-freeze-drying) [16] and other methods (e.g., cogrinding, adsorption on solid carrier) [17,18,19,20,21]. The materials produced by these methods result in an amorphous structure in most cases. Although solubility is better, thermodynamic stability is worse than that of the crystalline material and therefore, the material becomes crystalline over time [22]. One way to increase the solubility [23,24,25,26] and bioavailability [27,28,29] of crystalline, poorly water-soluble drugs is the application of a drug delivery system (DDS), with the use of different carriers, such as liposomes, polymer particles, dendrimers, miscellaneous nanoparticles, or the immobilization on an inorganic carrier, e.g., on a mesoporous silica material (MSM) [30,31,32]. MSMs are widely used, due to their numerous advantages, including a large surface area and pore volume, well-defined pore size, nontoxicity and biocompatibility. They have silanol groups on their surface, which are easy to modify. The crystallization of amorphous drugs that are insoluble or poorly soluble in water may be reduced in MSMs within the mesoporous space [33]. This reduces the lattice energy of the amorphous drug compared to the crystalline drug and results in an improved dissolution rate, as well as increased bioavailability [34].

There is a need to develop a formulation that is capable of a faster effect without increasing side effects. Increasing the bioavailability of metronidazole by immobilization on silica seems to be feasible. Yunessnia Iehi and coauthors [35] loaded MTZ on crystalline self-developed SBA-15 nanowhiskers, while Liu et al. [36] used mesocellular foam silica (MCF). Apart from them, hardly any indication has been found in the literature that the suitability of SiO_2_ for this drug has been investigated. In the fixation experiments on MTZ mesoporous silica materials, SBA-15, [37] SBA-15 with a modified surface [35], MCM-41 [38,39], MCM-41 with modified surfaces [40], mesoporous silica foam (MSF) [36] and mesoporous silica gel (MSNgel) with different pore sizes [41] were used. They tried to increase the loading efficiency by modifying the surface and increasing the pore size, which in the best case reached only 71.4%. In the case of functionalized surfaces, the fixation efficiency may decrease, and due to the electrostatic interaction between MSM and MTZ, the release may be significantly modified [40]. Aqueous solutions have been used to produce MTZ-containing MSM, usually with magnetic stirring or standing for 24 h [37,38,40,41]. Liu at al mentioned that the mixing of MTZ and MSF started with the use of ultrasonication, but continued with magnetic stirring [36]. In our experiment, we looked at the effect of the MTZ concentration and MTZ/MSM (MTZ to Syloid and SBA-15) concentration ratio, as well as ultrasound and organic solvent, on the fixation efficiency. We aimed to investigate this approach in more detail, and the efficiency of metronidazole fixation, the most important physical parameters of the prepared metronidazole on mesoporous silica, the critical physical parameters of the prepared metronidazole immobilized on the carrier, as well as the releasing behavior of metronidazole from the DDS.

## 2. Materials and Methods

### 2.1. Materials

Metronidazole was obtained from Aarti Drugs Limited (Mumbai, India) with European Pharmacopoeia (EuP) specification. High-purity water (18.3 MΩ/cm) was produced by Millipore equipment (Merck Millipore, Burlington, Massachusetts, USA). Syloid^®®^72FP was from Grace Materials Technologies Ltd. (Columbia, SC, USA). The parent silica SBA-15 (Santa Barbara Amorphous type material) material was synthesized according to the well-known procedures of Szegedi et al. [42]. Ethyl acetate was purchased from Merck KGaA (Darmstadt, Germany).

### 2.2. Preparation of Metronidazole Immobilized on Mesoporous Material

For the experiments, 1 *w*/*v*% (1 g MTZ in 100 mL water) and 3 *w*/*v*% (3 g MTZ in 100 mL ethyl acetate) of solutions were prepared from metronidazole.

Three different techniques were employed to immobilize MTZ from an aqueous solution on MSMs. First, incipient wetness impregnation was used by allowing the carrier to be just wetter [43]. One gram of silica was impregnated with as much aqueous solution of MTZ per 500 µL as the material was able to absorb. This was 3 mL for Syloid (sample *SyM1*) and 5 mL for SBA (sample SM1). The materials were lyophilized. In the next impregnation technique, 1 g of the particular carrier materials was stirred with a 1 wt% MTZ solution for 2.5 h and then lyophilized (samples *SyM2* and *SM2*). In the third technique, the material was filtered off and lyophilized (samples *SyM3* and *SM3*) after 2.5 h of mixing and stirring the carrier with the solution.

To increase efficiency, we used ultrasonication (US) for 24 h with a solution containing double and fourfold amounts of MTZ (samples *SM4* and *SM5*) instead of stirring (sample *SM3*), and it was repeated with a solution containing four times the amount of MTZ (sample *SM5*). Impregnation tests were also performed with an organic solvent [44]. MSM was stirred or subjected to ultrasonic agitation in an ethyl acetate solution (1.5 wt%) for 3 days (sample *SM6* and sample *SM7*, respectively). Metronidazole immobilized on the mesoporous material was filtered off from the solutions and lyophilized.

The most important parameters for the preparation of MTZ immobilized on two types of MSM are summarized in Table 1. The MTZ content was determined by spectrophotometry. Absorbance was read at 277 nm [45].

The lyophilizer was a Scanvac Coolsafe 55-9 Pro Control type 5-tray freeze dryer.

### 2.3. Low-Temperature Nitrogen Adsorption/Desorption

In order to characterize the porous texture of the samples, we performed the nitrogen adsorption isotherms at −196 °C with an Autosorb 1C (Quantachrome. Boynton Beach. FL, USA) static volumetric computer-controlled instrument by measuring the increase of volume at equilibrium as a function of relative pressure. All the samples were outgassed in vacuum at 60 °C prior to the gas adsorption measurements. Adsorption data were obtained with the use of a ca. 0.05 g sample and successive doses of the selected gas until p/p_0_ = 1 relative pressure was reached. The apparent surface area was calculated with the Brunauer–Emmett–Teller (BET) model [46]. Total pore volume (V_tot_) was calculated from the amount of nitrogen vapor adsorbed at a relative pressure p/p_0_ close to 1, assuming that the adsorbed N_2_ was in condensed form. We calculated the pore size distribution (PD) from the desorption branch of the isotherms, using the Barrett–Joyner–Halenda (BJH) method. The micropore volume (V_m_) was deduced from the Dubinin–Raduskevich (DR) model. The SSA coverage was calculated from the SSA ratio of the drug loaded carrier to the pristine carrier supposing that the decrease in the SSA could be attributed to the attachment of drug molecules to the surface.

### 2.4. X-ray Photoelectron Spectroscopy

X-ray photoelectron spectroscopy (XPS) was performed with a Kratos XSAM 800 spectrometer operating in the fixed analyzer transmission mode (pass energy 40 eV), using a Mg Kα1.2 (1253.6 eV) excitation. Survey spectra were recorded in the kinetic energy range of 150–1300 eV with 0.5 eV steps. Photoelectron lines of the main constituent elements (O1s, N1s, C1s and Si2p) were recorded with 0.1 eV steps. The spectra were referenced to the C1s line (binding energy: BE = 285.0 eV) of the hydrocarbon-type carbon. A quantitative analysis and layer thickness calculations were performed with the XPS MultiQuant 7.8 program [47], based on the peak area intensity after removal of the Shirley type background, with the use of the cross section data of Evans et al. [48] and asymmetry parameters of Reilman et al. [49]. A correction for surface contamination was performed with the method of Mohai [50].

### 2.5. Scanning Electron Microscopy (SEM)

The initial materials and prepared mesoporous materials were characterized by SEM (Zeiss EVO40) using a tungsten hairpin filament operated at 20 kV. The images were acquired at different magnification used a secondary electron detector (SED).

### 2.6. X-ray Diffraction Analysis

The X-ray powder diffraction analysis was performed with a Philips PW-1050 diffractometer, equipped with a Bragg–Brentano parafocusing goniometer. The Cu anode tube operated at 40 kV and 35 mA power, and the device had a secondary beam graphite monochromator and a proportional counter. The scans were recorded in step mode with a 0.04° step for 1 s between 10° and 70° 2θ angles.

### 2.7. Thermogravimetric and Differential Scanning Calorimetric (TG-DSC) Test

The thermal behavior of samples was investigated with a Setaram LabsysEvo (Lyon, France) TG-DSC system in a flowing (50 mL min^−1^), high purity (99.999%) argon atmosphere. Samples, without any sample preparation, were placed directly into 100 μL Al crucibles (the reference cell was empty) and were heated from 25 °C to 250 °C with a heating rate of 10 °C min^−1^. The obtained data were baseline-corrected and further processed with the thermoanalyzer’s processing software (Calisto Processing, ver. 2.06), in which the melting point (extrapolated onset temperature), peak maximum and enthalpy were determined by the baseline integration method. At the same time, the degree of crystallinity was calculated based on comparing the melting enthalpies of samples with the melting enthalpy of the fully crystalline starting metronidazole reference material. The thermal analyzer (both the temperature scale and calorimetric sensitivity) was calibrated with a multipoint calibration method. Seven different certified reference materials were used to cover the thermal analyzer’s entire operating temperature range.

### 2.8. Size Distribution

The particle size distribution was determined in an aqueous suspension after a 60 s sonication in the presence of a few drops of detergent, with a HORIBA Partica LA 950 A2 apparatus.

### 2.9. Study of Metronidazole Release from a Drug Delivery System Containing Silica

One gram of MTZ or an immobilized MTZ sample containing 1 g of MTZ was stirred in 100 mL of water at room temperature. At certain intervals (5, 10, 15, 30, 60 and 90 min), 5 mL of solution was withdrawn, and 5 mL water was added. The MTZ concentration of the extracted solution was determined by spectrophotometry. Absorbance was measured at 277 nm [38].

## 3. Results

### 3.1. Physicochemical Characterization of MTZ Immobilized on MSMs

#### 3.1.1. Nitrogen Physisorption Analysis

The adsorption–desorption isotherms for the pure and MTZ-immobilized carriers are shown in Figure 1. The isotherms of Syloid based samples (Figure 1A) were of type Iva, according to the IUPAC classification with H2 hysteresis loops, which refers to interconnected networks of pores of different size and shape in the mesoporous range. On the isotherms of SBA-15 based samples, the hysteresis loop (Figure 1B,C) was a typical H1 loop suggesting a narrow distribution of uniform, typically cylindrical pores. The pore structure mainly comprised wider mesopores and macropores, although micropores could also be detected in minor amounts (ca. 0.3 cm^3^g^−1^). Due to the differences in the pore structure, the specific surface area (SSA) of the two carriers was significantly different. The SSA of SBA15 (882 m^2^ g^−1^) was more than double that of Syloid (368 m^2^ g^−1^). Considering the almost identical total pore volume (1.1 cm^3^g^−1^) of the two carrier materials, the much higher SSA of SBA15 could be ascribed to its microporosity.

The major parameters obtained from the physisorption measurements of the starting MSM carriers and the materials containing MTZ are summarized in Table 2. It can be seen that the SSA and pore volume decreased as the active ingredient content increased (Table 1). After the immobilization test, both the SSA and the total pore volume (that mainly comprised mesopores) of Syloid were almost halved to ~198 m^2^g^−1^ and 0.6 cm^3^g^−1^, respectively, in the case of sample *SyM2*. Assuming that drugs attached to the surface reduced the SSA, we could calculate the surface coverage of the carrier from the decrease of the SSA values. The obtained value for *SyM2* was equivalent to a surface coverage of 46%, while the same value was a meager 16% for sample *SyM1* and 9% for sample *SyM3* (Table 3).

The total pore volume of the mesoporous (V_meso_) SBA-15 was significantly reduced in sample *SM2*, while only a slight decrease was detected in samples *SM1* and *SM3*. The specific surface area of sample *SM2* was greatly reduced from ~882 m^2^ g^−1^ to ~203 m^2^ g^−1^, indicating a surface coverage of 77%, while this value was 28% for sample *SM1* and 33% for sample *SM3*.

The above results indicate that the binding affinity of MTZ towards SBA-15 was higher than towards Syloid, despite the larger mean pore size of Syloid (11.3 nm) compared to SBA-15 (5.95 nm), determined from the desorption isotherms. To achieve complete drug loading, the pore diameter to drug molecule size ratio must be higher than one theoretically. However, in order for the pore to be properly accessible to the drug molecule, the ratio must preferably be above three [51]. The size of an MTZ molecule is about 1–1.5 nm, suggesting that it can fit in the mesopores of Syloid and SBA as well. It follows that the size of the micropores does not explain the great difference in the loading efficiency between the two MSMs. It can most probably be ascribed in part to the higher SSA of SBA-15 containing larger quantities of silanol groups that are favorable for the interaction with MTZ through hydrogen bonds [52,53,54,55]. This may be one of the most important parameters to achieve the highest drug loadings; therefore, further experiments for increasing the efficacy were only continued with SBA-15.

To increase the efficiency of carrier binding, we investigated an increased solute concentration, the effect of ultrasonic (US) agitation and the use of an organic solvent with SBA-15.

When MTZ dissolved in water was ultrasonically forced into the mesopores, the specific surface area of SBA-15 was reduced significantly from approximately 882 m^2^ g^−1^ to ~451 m^2^ g^−1^ (sample *SM4).* However, increasing the MTZ concentration (sample *SM5)* resulted in only a slight increase in efficiency according to the SSA (~428 m^2^ g^−1^) (Table 2). This shows that surface coverage was 49% or 51% (Table 3) showing an increase of approximatively 15–20% compared to previous tests. Ultrasonic agitation theoretically increases the interaction between phases with a local microturbulent flow and shockwaves that provide a better mixing, shearing in the solutions or suspensions [56]. The results suggested that while US agitation had a significant effect, the ratio of SBA-15 to the starting MTZ material was negligible. Although we were able to increase surface coverage by the application of ultrasound, the degree of binding did not increase considerably, despite the addition of more drugs to the system.

Increasing the binding efficacy by using an organic solvent is based on earlier findings claiming that active substances do not bind sufficiently to the surface when dissolved in a highly polar solvent such as DMSO or water [57]. One of the conditions for good immobilization on mesoporous silica and the establishment of a suitable interaction between the drug and the carrier is the use of a solvent which has a low relative permittivity but at least the same solubility for metronidazole as water [58,59]. MTZ is highly soluble in DMSO, methanol and ethanol, and their relative permittivity is only half that of water (80.5). However, ethyl acetate stands out with its low relative permittivity of 6.02 combined with a good MTZ solubility of 33.40 ± 1.71 mg mL^−1^ at 25 °C [10]. Therefore, we performed a few tests (samples *SM6* and *SM7*) with an ethyl acetate solution applying the combination of stirring and ultrasound agitation.

Ethyl acetate as a solvent reduced the specific surface area of the pores drastically (sample *SM7*). The adsorption isotherms in Figure 1C show that a significant portion of the mesopores of SBA-15 in samples *SM6* and *SM7* were filled with MTZ. For sample *SM6*, the pore coverage was 86%, while for sample *SM7* this value reached 91%.

There may be room for a further increase in the efficiency of drug loading, which is most often accomplished by a surface modification of the carrier or by changing the polarity of the system. To increase the binding of MTZ to MSM, Yunessnia lehi et al. previously modified SBA-15 nanowhiskers with tannin, achieving a capture efficiency of 71.4% [35]. However, with the right choice of solvent as well as mixing and ultrasound, we were able to achieve an 86% and 91% drug loading, respectively, without surface modification.

#### 3.1.2. Surface Characterization

Figure 2 shows typical X-ray photoelectron spectra of the MTZ adsorbed on silica. Besides the components of MTZ and the substrate, only a low level of carbonaceous contamination could be detected. The contamination was also low (lower than usual) on the precursors. After correction [50], their compositions were quite close to the expected theoretical values (Table 4. rows 1–2). The positions of the component lines were in good agreement with the expected chemical states of the materials.

Table 4 shows the surface composition of the adsorbed samples, calculated by the “*infinitely thick homogeneous sample*” model. Although this approach can give relative information on the adsorbed quantity of MTZ, the values are not accurate because the samples are not homogeneous.

To estimate the adsorbed amount of MTZ, we used the following model: the spherical particles of silica were covered by a carbonaceous contaminant layer followed by the metronidazole layer, as shown in Figure 3A’s inset. The contamination found on silica was equivalent to a 0.17 nm thick carbonaceous layer, whose thickness was kept constant. The thicknesses of MTZ layers were calculated with this *layers-on-sphere* model for each sample. The measured and simulated composition values are displayed in Figure 3A**,** as a function of the “equivalent layer thickness”.

The fitting of nitrogen (the marker element of MTZ) and carbon concentrations to the simulated data was almost excellent, while the small discrepancies of silicon and oxygen could be explained by the formation of OH groups during preparation or the nonspherical shape of the silica particles. Obviously, metronidazole did not form a uniform and continuous layer on the substrate but the “equivalent layer thickness” values could give approximations for the adsorbed quantity.

Of the Syloid samples, the surface nitrogen and thus the active ingredient concentration was the highest on sample *SyM2*, while it was very low on samples *SyM1* and *SyM3* (Table 4). The low drug content of sample *SyM1* shows that the carrier was able to bind only a small amount of liquid and therefore only a small amount of drug was used. Among the SBA-15 samples (samples *SM1, SM2* and *SM3*), prepared similarly to the Syloid samples (samples *SyM1, SyM2* and *SyM3*), sample *SM2* contained the largest level of active ingredient, while it was very low for samples *SM1* and *SM3*. Among the samples, samples *SM4* and *SM5* also contained very little active substance on the surface, while samples *SM6* and *SM7* contained the highest amount.

The calculated “equivalent layer thickness” values were in good agreement with other measured parameters, as shown in Figure 3B. The equivalent layer thickness was proportional to the total metronidazole content of the samples, measured during preparation (Table 1,) and was inversely proportional to the total pore volume (Table 3, Appendix A). These agreements suggest that most of the adsorbed material was located on the top surface and in shallow pores near to the surface. By a moderate estimate, based on the inelastic mean free path of electrons in silica, the pores involved in adsorption were not deeper than 9 nm.

#### 3.1.3. Size Distribution

The average size of Syloid samples was less than 10 µm. The size distribution of sample *SyM1* was the most similar to that of the drug carrier. For samples containing MTZ bound to the SBA-15 carrier, sample *SM7* showed the size distribution most similar to that of the reference carrier. Sample *SM6* fell in a slightly smaller size range, as the silica agglomerate seemed to disintegrate a little as a result of the ethyl acetate treatment and magnetic stirring. All other composites made in water as a reaction medium showed a little bit larger average size (Figure 4).

#### 3.1.4. SEM Studies

The morphology of the treated Syloid and SBA-15 samples were studied by SEM. SEM images of the samples of the Syloid series showed irregularly shaped highly agglomerated particles for both the pure and the MTZ-coated sample (Appendix A). According to the data sheet of the Syloid reference, the mean size of Syloid particles is 12 µm. However, our SEM investigations revealed that the size of the primary particles was typically much smaller than 1 µm, as can be seen in the high-magnification images (Figure 5A). These nanoparticles aggregated into larger particles of 5–7 µm in size (Figure S3), which was in good agreement with the laser-diffraction-based size distribution analysis (Figure 4). No significant difference could be revealed between the reference and the MTZ-coated Syloid particles. Even at high magnification, MTZ could not be clearly distinguished from the surface of Syloid (Figure 5A,B). However, the BET and XPS investigations supported that the MTZ was located on the surface as well as in the pores of the carriers.

The morphology of MTZ-treated samples on an SBA-15 carrier is shown in Appendix A**.** The morphological differences are obvious between the Syloid and SBA-15 carriers. In contrast to Syloid based samples, the SBA-15 samples were composed of particles with a more spherical shape and of similar size after lyophilization.

The different processing steps (e.g., stirring method, time) apparently did not affect the morphology of the products. Appendix A shows that the mean size of the primary particles was 2–4 µm for each sample, but they form agglomerates of ~10 µm, which was again in good agreement with the results of the size distribution analysis (Figure 4).

The effect of the MTZ treatment was not obvious for samples *SM1, SM4* and *SM5*; even at high magnification, we could not see evidence of the appearance of MTZ on the carrier’s surface. This was not totally surprising, since these samples contained a relatively low amount of MTZ, as can be seen in Table 1. Therefore, we assumed that MTZ was present in the pore structure characteristic of SBA-15. This was supported by the fact that the small amount of material bound by SBA significantly reduced the total volume of the micropores based on the adsorption measurements (Table 3), and XPS also showed that there was little active substance on the surface (Table 4).

However, very small particles, presumably MTZ, were noticeable on the surface of the samples *SM2, SM6* and *SM7,* where the latter two were prepared in ethyl acetate solutions with a high amount of MTZ. On sample *SM2*, MTZ was visible because the sample was unfiltered and the residual MTZ remained in the gaps between agglomerated particles (Figure 5C,D). This finding was consistent with the results of adsorption and XPS results obtained earlier (Table 3 and Table 4, Figure 3), which showed that the active substance covered most of the meso- and micropores and the surface.

#### 3.1.5. XRD Measurements

We used an X-ray diffraction (XRD) analysis (Figure 6) to monitor the changes in the studied carriers during the immobilization process.

Figure 6A shows the XRD patterns of the Syloid based samples. The figure includes the reference as well as the samples coated with MTZ applying different preparation conditions. The reference Syloid sample was seemingly amorphous, but samples *SyM1* and *SyM3* showed no sign of a crystalline structure of MTZ either. However, on the XRD pattern of sample *SyM2*, characteristic diffraction peaks of MTZ were clearly visible beside the amorphous peak of Syloid. This result could be explained on the basis of the synthesis conditions (Table 1), since for this sample, the product was not filtered at the end of the process and the residual MTZ remained in the sample. As mentioned earlier, according to Maleki et al., the ratio of the pore diameter to the drug molecule should be greater than three [51]. However, too large a ratio may cause the crystallization of the active ingredient [60]. The size ratio of the Syloid pore to the MTZ molecule is more than seven, so the large pore size of Syloid may leave enough room for MTZ to crystallize during storage [51]. However, considering the crystalline size (55 nm) of the MTZ determined by the Scherrer formula, it is also possible that the MTZ accumulated on the surface in larger fraction, showing the characteristic MTZ spectra.

Figure 6B shows the results of the XRD analysis of the samples with the SBA-15 carrier. Similar to Syloid, the SBA-15 carrier was also amorphous. Most of the samples prepared in an aqueous solution (samples *SM1, SM3, SM4* and *SM5*) were amorphous, too. The only exception was sample *SM2*, which was also prepared in an aqueous solution but was not filtered. Here, less intensive peaks of MTZ could be detected, and at the same time, the definite amorphous peak of SBA-15 flattened a little. The crystalline peak could be attributed to the MTZ accumulating on the surface of the sample in larger fractions.

For samples *SM6* and *SM7*, the change was quite significant in terms of crystallinity. In both cases, we observed the intense peaks corresponding to MTZ with a crystalline size of 90 and 141 nm, respectively. In these tests, ethyl acetate as solvent and a higher MTZ content were used compared to the samples made in an aqueous solution. Since the pore size of the SBA-15 carrier was the same in samples *SM6* and *SM7* as in samples *SM1, SM3, SM4* and *SM5,* and in all samples, the unbound active ingredient was filtered out, therefore the MTZ peaks were caused by the large amount of immobilized active substance in the mesopores and on the surface.

#### 3.1.6. Thermal Analysis

In the case of Syloid bound MTZ samples, thermal tests revealed that the MTZ active substance was in an amorphous form in samples *SyM1* and *SyM3*, while a small melting endotherm shifting towards lower temperatures (compared to the melting endotherm of the bulk MTZ) was visible (T_onset_: 112.78 °C, T_peak max._: 127.78 °C, *ΔH*_fus_: 19.26 J/g) in the case of sample *SyM2*. This means that the MTZ was in a crystalline form. The shift of the melting point to lower temperatures could also mean that the MTZ was in a nanocrystalline form. These findings were confirmed by the results of the XRD analysis too. In three of the four plotted samples in Figure 7A, namely samples *SyM1*, *SyM3* and the Syloid carrier, a small and broad endotherm is visible up to 100 °C, which is the result of the evaporation of a few percent of the physically bound water, which is confirmed by the mass loss on their corresponding mass loss (TG) curves (Appendix A).

Considering thermal properties, a similar trend to that in the case of the Syloid–MTZ composites described above could be observed in the case of SBA-15 bound MTZ samples. In most investigated samples (samples *SM1, SM2, SM3, SM4* and *SM5*), no melting endotherm characteristic of MTZ was observed in the measured temperature range, suggesting that the MTZ active component was present in amorphous form in the prepared composites. In contrast, samples *SM6* and *SM7* exhibited two endotherm peaks each: one smaller at lower temperatures (sample *SM6*: T_onset_: 109.81 °C, T_peak max._: 121.35 °C, *ΔH*_fus_: 17.61 J g^−1^; sample *SM7*: T_onset_: 108.97 °C, T_peak max._: 119.48 °C, *ΔH*_fus_: 12.95 J g^−1^) and one larger at higher temperatures (sample *SM6*: T_onset_: 155.63 °C, T_peak max._: 160.76 °C, *ΔH*_fus_: 64.79 J g^−1^; sample *SM7*: T_onset_: 155.91 °C, T_peak max._: 161.70 °C, *ΔH*_fus_: 91.94 J g^−1^). We assume that the two endotherm peaks suggested the presence of MTZ in different crystalline forms in the carrier. One fraction of MTZ was nanocrystalline, which melted at lower temperatures, while the larger, microcrystalline fraction of MTZ bound to the surface of the particles melted at higher temperatures. For reference, the heat flow curve of MTZ is also plotted in Figure 7B (MTZ: T_onset_: 157.61 °C, T_peak max._: 168.30 °C, *ΔH*_fus_: 201.55 J g^−1^). It can be clearly seen that the main parameters of the two larger endotherms (except the melting enthalpies, *ΔH*_fus_, for samples *SM6* and *SM7*) almost match the parameters of the bulk reference MTZ. The presence of crystalline MTZ (in samples *SM6* and *SM7*) was also confirmed by the results of the XRD analysis. The exothermic peaks above 250 °C corresponded to the thermal degradation of the MTZ.

According to the literature, as mentioned earlier, the mesoporous structure prevents the crystallization of the active ingredient in the pores and on the surface [61]. However, other researchers have already obtained similar results in the case of ibuprofen and fenofibrate, where active substances with crystalline structures were detected both in the pores and on the surface after the fixation on porous silica [62,63]. This was explained by the large size of the pores compared to the molecular size; their ratio was greater than 20 in both cases [64]. For our samples, this cannot explain the crystalline state. It is far more likely that it is due to a large amount of material on the surface, as shown by the nitrogen adsorption and XPS studies (Table 3 and Table 4).

### 3.2. In Vitro Release of MTZ from MSM

We performed release studies on four samples with the highest drug content (samples *SyM2, SM2, SM6* and *SM7*) to evaluate the bioavailability of the drug immobilized on the silica carriers. Figure 8A shows that the dissolution profiles were similar for all immobilized drugs, while different for the micron-size MTZ particles. Metronidazole showed a saturation curve reaching a dissolution degree of 81.8% in 90 min. The dissolution of MTZ did not reach 100% due to the applied conditions, such as concentration, temperature and time. The immobilized drugs reached a higher ratio as compared to the dissolution of MTZ, around 90%. Sample *SM7* had the best release (96.1%), followed by sample *SM6* (90.0%) and sample *SM2* (89.7%), and sample *SyM2* was the last with 86.8%. These results are in line with the literature, according to which, poorly water-soluble active substances, such as MTZ, dissolve and are released faster after immobilization on mesoporous silica than the original active substance itself dissolves [65]. Maleki et al. immobilized itraconazole on SBA-15 and found that twice as much itraconazole dissolved as the pure drug itself in 2 h [51]. In our experiment, the difference was not so large; only a 5–10% difference could be seen.

There were no significant differences in the dissolution rate of MTZ after immobilization in the particular tests, even though no filtration was applied to samples *SyM2* and *SM2* to remove unbound active substance. For each sample, a rapid dissolution was followed by a slow release, which is typical, since it takes place in two steps. The first is a quick dissolution from the surface and then a slower release from the deeper pores [62,66,67]. In this regard, the shape of the curves confirmed that a substantial part of the MTZ was situated in the pores of the carrier.

There was not much difference between the dissolution of MTZ immobilized on either the Syloid or SBA-15 carriers within the first 5 min. The dissolution of the surface-bound drug from samples *SM2, SM6* and *SM7* of SBA-15 were similar and the drug release was above 50%, reaching 60% in all cases within 5 min. For the sample *SyM2* of Syloid, this parameter was the lowest with 54.8%. These results are similar to those found in the literature, where 60–80% of itraconazole bound to SBA-15 was dissolved within 5 min [68]. Both Syloid and SBA-15 have silanol groups on the surface, which bind the drug to the mesoporous materials with weak hydrogen bonds [69]. This surface-bound drug dissolves rapidly. Due to its smaller SSA with fewer silanol groups, Syloid can bind less active ingredient. After this fast dissolution, a further dissolution and release of the additional drug slow down, as the active substance must escape from the deeper pores. This process is determined by the rate of slow diffusion, which depends on pore size [70]. The larger the pore size, the faster the drug can escape from the pores and diffuse into the solution [51]. For the MSMs used here, the average mesopore size of Syloid was 11.3 nm, while it was 5.95 nm for SBA-15, which was about a twofold difference, so the release from Syloid could theoretically be faster. This was also supported by the results of the experiment, if we looked at the percentage of MTZ dissolution at certain times (Appendix A). The dissolution of MTZ from the Syloid sample *SyM2* in 5 min and 15 min showed a greater difference (2.5%) than from sample *SM2* (1.1%). The dissolution of the active agent from sample *SM2* after 5 min showed only a 1.1% increase by 10 min, but then a steady increase in dissolution until 60 min (4.5%, 8.0% and 10.2%, respectively) was observed. The dissolution of sample *SM7* showed a similar trend in the same time period, with slightly higher values (4.8%, 8.2%, 8.6% and 10.2%). In the case of sample *SM6*, the percentage of drug released by diffusion increased only until 15 min (2.5%, 17.0%) and then decreased until 90 min (8.3, 2.5 and 1.0%).

If we look at these dissolution values and dissolution speeds (Appendix A), we can see that the velocity profiles were similar for the immobilized drug but different from the original MTZ. There were some differences in the speed of composites during certain time periods, but the trend was the same in each case (Figure 8B). Within the first 5 min, a rapid and almost uniform dissolution occurred from the surface, as previously noted. After this, there was a rapid decrease in speed until 10 min, as the drug ran out of the surface. The release and diffusion from the deeper pores reached their highest rate after 15 min, after which they decreased continuously.

## 4. Conclusions

In order to increase the apparent solubility of metronidazole, which is poorly soluble in water, we used immobilization on mesoporous carriers (Syloid and SBA-15) using different immobilization techniques (dripping, dipping, magnetic stirring, ultrasound and filtering) and solvents (water and ethyl acetate) to obtain mesoporous silica materials containing metronidazole. Materials produced in an aqueous medium contained little active ingredient, but the drug was in an amorphous form. In our systematic experiment, we proved that the loading efficiency could be further increased by ultrasound agitation and using a carefully chosen organic solvent. Using ethyl acetate and magnetic stirring, we achieved a surface coverage of 86% on SBA-15, while ultrasonic agitation improved coverage to 91%. Metronidazole was nanocrystalline in the pores of the silica, while some of the drug was microcrystalline on the surface of the silica. The dissolution of MTZ was 5–10% faster when immobilized onto a carrier than the solution of the microcrystalline drug.

## Figures and Tables

**Figure 1 pharmaceutics-14-02332-f001:**
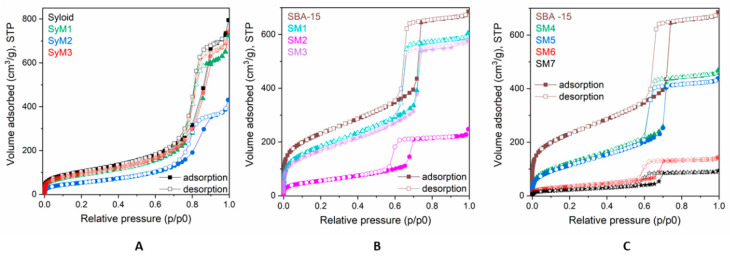
Adsorption isotherms of metronidazole–MSM materials; (**A**): Syloid samples, (**B**) and (**C**): SBA-15 samples.

**Figure 2 pharmaceutics-14-02332-f002:**
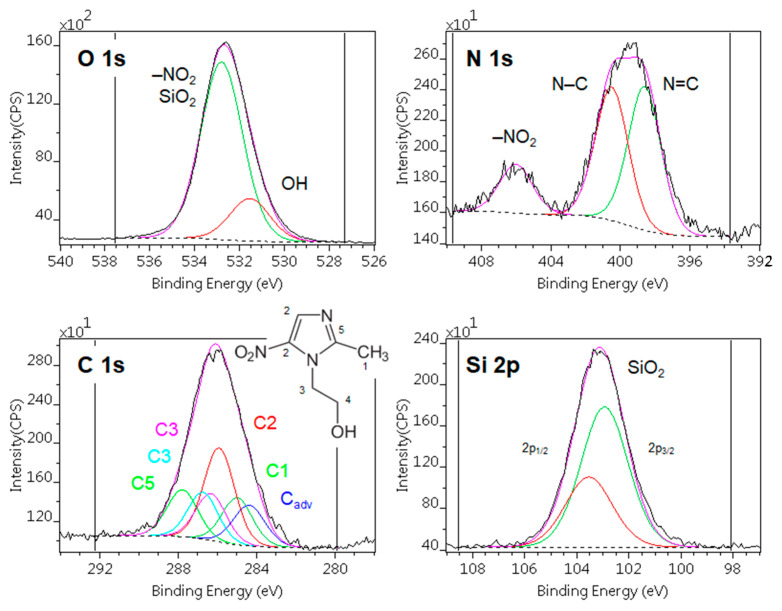
Typical X-ray photoelectron spectra of the metronidazole adsorbed on silica (sample *SyM2*). Photoelectron lines are decomposed to various chemical states.

**Figure 3 pharmaceutics-14-02332-f003:**
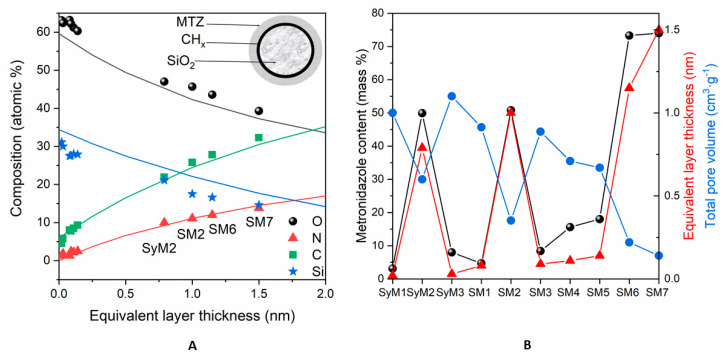
(**A**): Surface chemical composition (atomic %) of the samples containing MTZ adsorbed on silica, calculated with the infinitely thick homogeneous sample model. Continuous lines are simulated data, while dots are the measured samples. Abscissae are the equivalent layer thickness (nm) of metronidazole, calculated by the layers-on-sphere model. (**B**): Comparison of the active agent content (mass %) of the samples with the corresponding equivalent layer thickness (nm) and total pore volumes (cm^3^ g^−1^).

**Figure 4 pharmaceutics-14-02332-f004:**
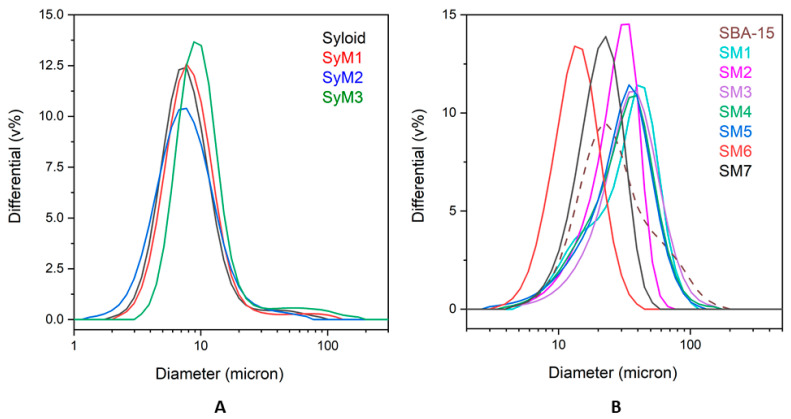
Size distribution of metronidazole–MSM particles ((**A**): Syloid, (**B**): SBA-15).

**Figure 5 pharmaceutics-14-02332-f005:**
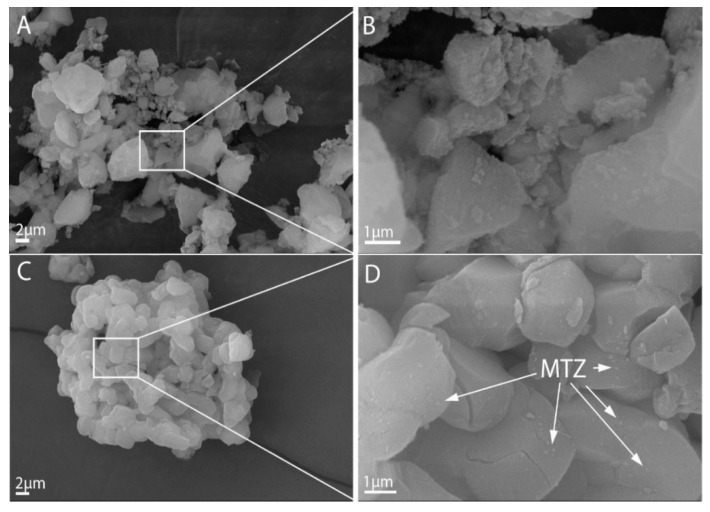
SEM images of the Syloid and SBA-15 (**A**,**C**) carriers and metronidazole immobilized on their surface (**B**,**D**).

**Figure 6 pharmaceutics-14-02332-f006:**
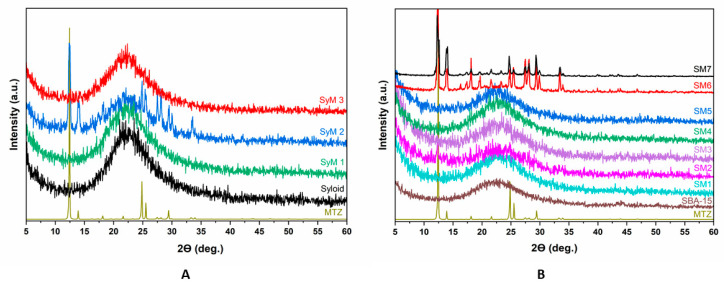
X-ray diffractograms of metronidazole–MSM particles ((**A**): Syloid, (**B**): SBA-15).

**Figure 7 pharmaceutics-14-02332-f007:**
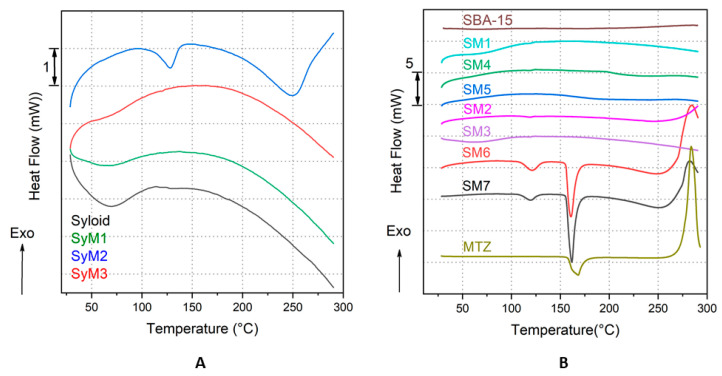
Heat flow (DSC) curves of the Syloid bound MTZ samples and the reference carrier, Syloid (**A**) and of the SBA-15 bound MTZ samples and the references SBA15 and MTZ (**B**).

**Figure 8 pharmaceutics-14-02332-f008:**
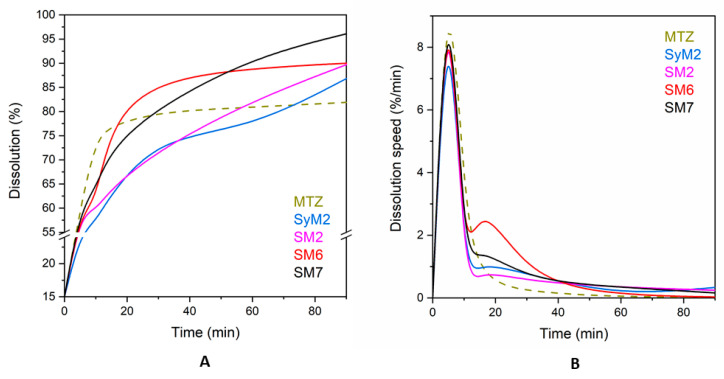
Dissolution of MTZ from its pure and immobilized form (**A**) and dissolution speed of MTZ from its pure and immobilized form (**B**).

**Table 1 pharmaceutics-14-02332-t001:** Parameters for the preparation of metronidazole immobilized on two types of MSM.

Sample	Type of MSM	MTZ Solvent	Ratio of MTZ to MSM	Method Applied					MTZ Content (%)
				IWI	Stir.	US	Time (h)	Filt.	
*SyM1*	Syloid	W	1:33	x					3.1
*SyM2*		W	1:2		x		2.5		49.9
*SyM3*		W	1:2		x		2.5	x	8.0
*SM1*	SBA	W	1:20	x					4.7
*SM2*		W	1:2		x		2.5		50.8
*SM3*		W	1:2		x		2.5	x	8.4
*SM4*		W	1:1			x	24	x	15.6
*SM5*		W	2:1			x	24	x	18.0
*SM6*		E	4:1		x		72	x	73.3
*SM7*		E	4:1			x	72	x	74.0

MSM: mesoporous silica material, MTZ: metronidazole, SM: metronidazole on SBA mesoporous silica material, SyM: metronidazole on Syloid mesoporous silica material, IWI: incipient wetness impregnation, Stir.: stirring, Filt: filtration, US: ultrasound, W: water, E: ethyl acetate.

**Table 2 pharmaceutics-14-02332-t002:** Data derived from N_2_ gas adsorption/desorption measurement.

Sample	V_tot_ (cm^3^ g^−1^)	PD (nm)	V_meso_ (cm^3^ g^−1^)	SSA (m^2^ g^−1^)
Syloid	1.13	11.3	1.1	368
*SyM1*	1.0	11.3	1.0	315
*SyM2*	0.6	10.2	0.6	198
*SyM3*	1.1	11.0	1.1	336
SBA-15	1.1	5.95	0.94	882
*SM1*	0.91	5.95	0.85	636
*SM2*	0.35	5.0	0.35	203
*SM3*	0.89	5.9	0.84	593
*SM4*	0.71	5.4	0.70	451
*SM5*	0.67	5.4	0.66	428
*SM6*	0.22	5	0.21	126
*SM7*	0.14	5	0.14	82

**Table 3 pharmaceutics-14-02332-t003:** Pore coverages of MTZ immobilized on the mesoporous silica materials.

Sample	Covered Total Pore Volume (%)	Covered Specific Surface Area (%)
*SyM1*	15	14
*SyM2*	47	46
*SyM3*	3	9
*SM1*	17	28
*SM2*	68	77
*SM3*	19	33
*SM4*	36	49
*SM5*	39	51
*SM6*	80	86
*SM7*	87	91

**Table 4 pharmaceutics-14-02332-t004:** Surface chemical composition (*atomic %*) of the metronidazole (MTZ) and Syloid carrier and the adsorbed MTZ samples, calculated by the *infinitely thick homogeneous sample* model. The last column shows the equivalent layer thickness (nm) of MTZ, calculated by the *layers-on-sphere* model.

Sample	O	N	C	Si	Equivalent Layer Thickness
Syloid *	67.3			32.7	
*SyM1*	63.1	1.3	4.5	31.1	0.00
*SyM2*	47.0	10.0	21.9	21.1	0.79
*SyM3*	62.4	1.8	5.8	30.0	0.03
*SM1*	63.2	1.3	8.0	27.5	0.08
*SM2*	45.7	11.1	25.8	17.5	1.00
*SM3*	62.2	2.4	7.8	27.6	0.09
*SM4*	61.2	2.2	8.5	28.0	0.11
*SM5*	60.3	2.5	9.3	27.9	0.14
*SM6*	43.6	12.0	27.8	16.6	1.15
*SM7*	39.3	13.8	32.3	14.6	1.50

* Corrected for carbonaceous contamination.

## Data Availability

Not applicable.

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
