# Peer review of "Immobilization of Metronidazole on Mesoporous Silica Materials"

_pharmaceutics, 2022, doi:10.3390/pharmaceutics14112332_

Round 1

Reviewer 1 Report

In this manuscript entitled "Immobilization of metronidazole on mesoporous silica materials", Szentmihályi and coworkers report the immobilization of metronidazole (MTZ) into mesoporous silica. The working hypothesis is that this approach reduces the size of the drug and improves its efficacy. Two mesoporous silica materials, Syloid and SBA-15, were used as the hosts to immobilize MTZ under several immobilization conditions. The immobilized drug was characterized, and the drug dissolution test was conducted. In general, this is a systematic study, however the novelty and significance level of this work is not clear. There are some other questions for the authors to consider, see details below. I suggest major revisions before I can recommend acceptance.

1)     The identified medical problems and the justification of research question is not clear to me. It is stated that "In the case of topical application in cosmetic preparations, … a number of side effects are expected. To reduce these symptoms, it is necessary to reduce the amount or size of the active substance.…". While I agree the size of drug could affect bioavailability and toxicity, is there any evidence that reducing the size / increasing the bioavailability could reduce the toxicity and side effects? Why not increased drug solubility leads to enhanced toxicity / side effects?

2)     MTZ loaded in mesoporous silica materials has been reported. Research gaps and new knowledge gained from this study should be clearly stated. Currently, it is not clear at all.

3)     Please provide TEM images / small angle XRD patterns of two MSMs. The pore structure has impact on the loading and release of guest molecules. Information such as on the pore length (if one dimensional pore channels exist) could be used in the discussion of Figure 3, page 10 (e.g., line 320-326, the pre depth involved in drug adsorption).

4)     The pore sizes should be determined for all samples. Does the pore size reduce after loading? Is it dependent on loading conditions?

5)     Lines 223-224, what is the unit of 1 / 3, nm?

6)     How did you calculate the pore volume / SSA coverage?

7)     What is the evidence of hydrogen bonding between MTZ and Si-OH?

8)     Discussion on the XRD results of SyM2 in Figure 6A, why not calculating the crystalline domain size of MTZ to provide more relevant information? Please compare this information to the pore size change / DSC results, then draw meaningful conclusions.

9)     Some minor grammar mistakes to correct, eg., ml should be mL; Table 1, MZT is wrong; surface area should be specific surface area in line 70; "mezoporous" at line 191 should be  "mesoporous". Please check the manuscript throughout.

Author Response

We appreciate, indeed very much the Reviewer’s effort she/he put into the evaluation of our manuscript and especially she/he valuable suggestions for the improvement of this work. Below we answer the Reviewer’s comments:

Reviewer 1:

1) The identified medical problems and the justification of research question is not clear to me. It is stated that "In the case of topical application in cosmetic preparations, … a number of side effects are expected. To reduce these symptoms, it is necessary to reduce the amount or size of the active substance.…". While I agree the size of drug could affect bioavailability and toxicity, is there any evidence that reducing the size / increasing the bioavailability could reduce the toxicity and side effects? Why not increased drug solubility leads to enhanced toxicity / side effects?

Answer:

It is a common assumption and stands to reason indeed that lower amount of drug reduces toxicity (or at least the probability of the toxicity) and side effects. In order to use less amount of drug with achieving the same effect, we need to increase its bioavailability. In our work, we increased the bioavailability and solubility of the drug by immobilizing it on a carrier. However, it was not the scope of this work to examine whether or not the increased bioavailability reduces toxicity or side effects in real. On the other hand, the increased solubility owing to either size reduction or immobilization of the drug is only apparent, the solubility does not actually change, only the surface area increases as the particle size decreases, which enables greater contact between the substance and the solvent.*

* Sun, J.; Wang, F.; Sui, Y.; She, Z.; Zhai, W.; Wang, C.; Deng, Y. Effect of particle size on solubility, dissolution rate, and oral bioavailability: evaluation using coenzyme Q10 as naked nanocrystals. Int. J. Nanomedicine 2012, 7, 5733–574, https://doi.org/10.2147/IJN.S34365

2) MTZ loaded in mesoporous silica materials has been reported. Research gaps and new knowledge gained from this study should be clearly stated. Currently, it is not clear at all.

Answer: Mesoporous silica materials as potential drug carriers have been investigated indeed using various model drugs. However, specifically for immobilization of MTZ on mesoporous silica only a few publications are available: SBA-15 [37], SBA-15 with a modified surface [35], MCM-41 [38,39], MCM-41 with modified surfaces [40], mesoporous silica foam (MSF) [36] and mesoporous silica gel (MSNgel) and MSNgel with different pore sizes [41]. They tried to increase the loading efficiency by modifying the surface and increasing the pore size, which in the best case reached only 71.4% [36]. In the case of functionalized surfaces, the fixation efficiency may decrease, and due to the electrostatic interaction between MSM and MTZ, the release may be significantly modified [40]. Aqueous solutions have been used to produce MTZ-containing MSM, usually with magnetic stirring or standing for 24 hours [37,38,40,41] One article mentioned that the mixing of MTZ and MSF started with the use of ultrasound, but continued with magnetic stirring [36]. In our experiment, we examined the effect of MTZ concentration and MTZ-MSM (MTZ to Syloid and SBA-15) concentration ratio, as well as ultrasound and organic solvent on the fixation efficiency.

  1. Yunessnia lehi, A.; Shagholani, H.; Nikpay, A.; Ghorbani, M.; Soleimani lashkenari, M.; Soltani, M. Synthesis and modification of crystalline SBA-15 nanowhiskers as a pH-sensitive metronidazole nanocarrier system. Int. J. Pharm. 2019, 555, 28–35, doi:10.1016/J.IJPHARM.2018.11.034.
  2. Liu, T.; Wang, K.; Jiang, M.; Wan, L. Interactions between mesocellular foam silica carriers and model drugs constructed by central composite design. Colloids Surfaces B Biointerfaces 2019, 180, 221–228, doi:10.1016/J.COLSURFB.2019.04.055.
  3. Chamorro-Petronacci, C.M.; Torres, B.S.; Guerrero-Nieves, R.; Pérez-Sayáns, M.; Carvalho-de Abreu Fantini, M.; Cides-da-Silva, L.C.; Magariños, B.; Rivas-Mundiña, B. Efficacy of Ciprofloxacin, Metronidazole and Minocycline in Ordered Mesoporous Silica against Enterococcus faecalis for Dental Pulp Revascularization: An In-Vitro Study. Materials (Basel). 2022, 15, 2266, doi:10.3390/ma15062266.
  4. Czarnobaj, K.; Prokopowicz, M.; Sawicki, W. Formulation and In Vitro Characterization of Bioactive Mesoporous Silica with Doxorubicin and Metronidazole Intended for Bone Treatment and Regeneration. AAPS PharmSciTech 2017, 18, 3163–3171, doi:10.1208/s12249-017-0804-3.
  5. Altememy, D.; Jafari, M.; Naeini, K.M.; Alsamarrai, S.; Khosravian, P. In-vitro Evaluation of Metronidazole loaded Mesoporous Silica Nanoparticles Against Trichomonas Vaginalis. Int. J. Pharm. Res. 2020, 12, 2773–2780, doi:10.31838/ijpr/2020.SP1.400.
  6. Czarnobaj, K.; Prokopowicz, M.; Greber, K. Use of Materials Based on Polymeric Silica as Bone-Targeted Drug Delivery Systems for Metronidazole. Int. J. Mol. Sci. 2019, 20, 1311, doi:10.3390/ijms20061311.
  7. Wang, W.; Wang, X.; Li, L.; Liu, Y. Anti-Inflammatory and Repairing Effects of Mesoporous Silica-Loaded Metronidazole Composite Hydrogel on Human Dental Pulp Cells. J. Healthc. Eng. 2022, 2022, 1–9, doi:10.1155/2022/6774075.

We inserted additional sentences into the introduction about it to make it clear for the readers.

3) Please provide TEM images / small angle XRD patterns of two MSMs. The pore structure has impact on the loading and release of guest molecules. Information such as on the pore length (if one dimensional pore channels exist) could be used in the discussion of Figure 3, page 10 (e.g., line 320-326, the pre depth involved in drug adsorption).

Answer: During characterisation we have performed TEM and SAXS measurements of the starting materials and one of each MTZ loaded samples. SAXS was suitable to show us the ordered arrangement of the pores in SBA-15 but not for determining the length of the individual pores. In case of Syloid the interconnected porous structure made calculation almost impossible. The TEM image exhibits a clear internal structure of prepared SBA-15. A 1-D porosity channels aligned parallel to one another with pore sizes of approximately 6 nm, which is close to the result of 5.95 nm calculated from N2 adsorption isotherm. We did not receive any new information from these methods compared to the isotherm measurement, in fact, less. That is why we did not put them into the manuscript. If the reviewer insists on it, we can put it in the manuscript, but wouldn't add anything new to the results.

4) The pore sizes should be determined for all samples. Does the pore size reduce after loading? Is it dependent on loading conditions?

Answer:  Pore sizes were determined for all samples and inserted in Table 2. The pore size, however, only slightly reduced after loading. We assume that substantial part of the pores were blocked by the drug molecules and instead of reducing the pore size they reduced the open porosity.

5) Lines 223-224, what is the unit of 1 / 3, nm?

Answer: These numbers represent the size ratios of pore diameter to drug molecule according to the previous sentence (nm/nm), numbers without a unit.

6) How did you calculate the pore volume / SSA coverage?

Answer: SSA coverage was calculated from the SSA ratio of the drug loaded carrier to the pristine carrier supposing that the decrease in the SSA can be attributed to the attachment of drug molecules to the surface. A comment for the calculation method is now inserted into the text.

7) What is the evidence of hydrogen bonding between MTZ and Si-OH?

Answer: Based on the literature, the OH groups of the mesoporous silica bind the drug molecule by hydrogen bonding as it was mentioned references 52-55. Furthermore Czarnobaj et al. used MCM-41 to bind MTZ and showed that MCM-41 binds MTZ by hydrogen bonds, since the MTZ molecule has 4 hydrogen acceptor sites and one proton donor site. Now, this reference is also inserted.

Czarnobaj, K.; Prokopowicz, M.; Sawicki, W. Formulation and in vitro characterization of bioactive mesoporous silica with doxorubicin and metronidazole Intended for bone treatment and regeneration. AAPS Pharm. Sci. Tech. 2017, 18, 3163-3171, DOI: 10.1208/s12249-017-0804-3

8) Discussion on the XRD results of SyM2 in Figure 6A, why not calculating the crystalline domain size of MTZ to provide more relevant information? Please compare this information to the pore size change / DSC results, then draw meaningful conclusions.

Answer: We calculated the crystalline domain size, however, peaks most probably are originating from the powders situating on the outer surface of the carrier and in this regard there is no connection with the pore size of the carrier. Molecules get into the pores remained in amorphous form. We supplemented and modified the manuscript with this new information and some comments about it.

9) Some minor grammar mistakes to correct, eg., ml should be mL; Table 1, MZT is wrong; surface area should be specific surface area in line 70; "mezoporous" at line 191 should be  "mesoporous". Please check the manuscript throughout.

Answer: We looked through the manuscript again and have checked grammar mistakes. Considering the abbreviation of millilitre we just used SI convention that says unit symbols abbreviating the name of a person start with a capital letter. On the other hand, reviewers are always right, so we corrected those units too.

Once more, we would like to thank for your valuable remarks and suggestion which undoubtedly enhances the scientific value of our manuscript.

Reviewer 2 Report

The work done by Klára Szentmihályi and Szilvia Klébert mainly investigated the fixation ability of SBA-15 and Syloid silica materials to MTZ molecules under diverse environments. They have found that the SBA-15 silica materials showed higher immobilization efficacy compared with Syloid materials. Furthermore, they found that the solvent species and ultrasonic treatment had influence on immobilization efficacy of MTZ molecules. Generally, this work is interesting and worth to be studied. This work can be considered published in Pharmaceutics journal after a minor revision. Before final acceptance, following issues should be solved.

Question 1: The authors claimed in the abstract as “another size reduction method is attachment of the drug molecules to a mesoporous carrier”. Where is the “another size reduction methods”? I cannot understand this method from the abstract.

Question 2: There are some errors such as spelling error. In addition, in page 5, line 203 is Table 2 but the authors marked Table 1. Please revise it.

Question 3: Despite the XPS results of sample SyM2 product including O, N, C, and Si elements being provided as shown in Figure 2, more detailed information should be supported.

Question 4: Please explain why the free MTZ release rate is only 80%.

Question 5: Please illustrate Figure 4 in detail and provide a clear picture.

Question 6: The language should be further polished before submission.

Question 7: as the author mentioned, the ultrasonic can influent the immobilization efficacy of drug. Comments?

To be brief, I suggested that the authors should clearly answer the above questions before accepting it for publication in Pharmaceutics journal.

Author Response

We appreciate, indeed very much the Reviewer’s effort she/he put into the evaluation of our manuscript and especially she/he valuable suggestions for the improvement of this work. Below we answer the Reviewer’s comments:

Reviewer 2:

Question 1: The authors claimed in the abstract as “another size reduction method is attachment of the drug molecules to a mesoporous carrier”. Where is the “another size reduction methods”? I cannot understand this method from the abstract.

Answer: We did not write such sentence in the abstract or in the whole manuscript. What we wrote like: “A relatively new method of size reduction is attachment of the drug molecules to a mesoporous carrier”

Question 2: There are some errors such as spelling error. In addition, in page 5, line 203 is Table 2 but the authors marked Table 1. Please revise it.

Answer: We hired a professional linguist to improve the language and correct all errors, including spelling ones, too. In line 203, we referred to Table 1 correctly, as the contents of the active ingredients are illustrated in that Table, while Table 2 shows some parameters determined from physisorption measurements.

Question 3: Despite the XPS results of sample SyM2 product including O, N, C, and Si elements being provided as shown in Figure 2, more detailed information should be supported.

Answer: The Reviewer did not specify what information he/she missed. So, we inserted an additional Table in the supplementary about the content of carbon atoms in different chemical bonds.

Question 4: Please explain why the free MTZ release rate is only 80%.

Answer: MTZ is a poorly soluble compound in water. In the experiment the room was air conditioned to 20 °C, in which MTZ with an initial concentration of 1g/100 ml (10mg/10mL) would have needed more time to fully dissolve than the duration of the experiment itself, since this was not a saturated solution at the beginning of experiment.

At a higher temperature, the dissolution of MTZ would certainly have reached 100%, however, in this case, the active ingredient would have dissolved faster from the surface of the mesoporous material, providing less information about the dissolution and diffusion processes.

Question 5: Please illustrate Figure 4 in detail and provide a clear picture.

Answer: We improved Figure 4 to have a more visible and clear picture of the results.

Question 6: The language should be further polished before submission.

Answer: We hired a professional linguist to improve the language.

Question 7: as the author mentioned, the ultrasonic can influent the immobilization efficacy of drug. Comments?

Answer:

We added the comment below about the positive effect of US to the discussion part:

Ultrasonic agitation increases the interaction between phases with local microturbulent flow and shockwaves that provides better mixing, shearing in the solutions or suspensions.

* Kwiatkowska, B.; Bennett, J.; Akunna, J.; Walker, G.M.; Bremner, D.H. Stimulation of bioprocesses by ultrasound. Biotechnol. Adv. 2011, 29, 768–780, https://doi.org/10.1016/j.biotechadv.2011.06.005

** Sun, T.;Dong, Z.; Wang, J.; Huang, F.H. ; Zheng, M.M. Ultrasound-assisted interfacial immobilization of lipase on hollow mesoporous silica spheres in a Pickering emulsion system: A hyperactive and sustainable biocatalyst. CS Sustainable Chem. Eng. 2020, 8, 46, https://doi.org/10.1021/acssuschemeng.0c06271.

*** V.C. Badgujar, K.C. Badgujar, P.M. Yeole, B. M. Bhanage, Investigation of effect of ultrasound on immobilized C. rugosa lipase: Synthesis of biomass based furfuryl derivative and green metrics evaluation study. Enzyme Microb. Technol. 2021, 144, 109738, https://doi.org/10.1016/j.enzmictec.2020.109738.

Once more, we would like to thank for your valuable remarks and suggestion which undoubtedly enhances the scientific value of our manuscript.

Round 2

Reviewer 1 Report

The authors have made appropriate changes. I recommend acceptance as it is.